# Experimental Granulometric Characterization of Wood Particles from CNC Machining of Chipboard

Pavol Koleda [1,*] , Peter Koleda [1] , Mária Hrčková [1] , Martin Júda [2] and Áron Hortobágyi [1]

[1] Faculty of Technology, Technical University in Zvolen, Študentská 26, 960 01 Zvolen, Slovakia; peter.koleda@tuzvo.sk (P.K.); hrckova@tuzvo.sk (M.H.); a.hortobagyi.a@gmail.com (Á.H.)
[2] Faculty of Wood Sciences and Technology, Technical University in Zvolen, T. G. Masaryka 24, 960 01 Zvolen, Slovakia; xjuda@is.tuzvo.sk
[*] Correspondence: pavol.koleda@tuzvo.sk

**Abstract:** The aim of this paper is to determine the particle size composition of the wood particles obtained from CNC milling the chipboard using an experimental optical granulometric method. Composite materials (chipboard) are the most-used materials in the woodworking and furniture industries. The proposed optical method of measuring particles' dimensions is compared to the sieving technique. The researched experimental method allows for the determination of not only the size of the fraction of an individual particle's fraction but also more detailed information about the analyzed wood dust emission, for example, the largest and smallest dimension of each single particle; its circularity, area, perimeter, eccentricity, and convex hull major and minor axis length; or the color of the particle.

**Keywords:** wood particles; optical analysis; MATLAB

## 1. Introduction

Machining is one of the highly used processes in modern-day industrial applications, with increasing demands from customers all over the world in areas such as transportation, medicine, surgery, automobiles, space, aeronautics, etc. [1]. The development of industry and the economy puts pressure on the speed of production for products and goods. In addition to the advantages, this trend also has many negative aspects, for example, the excessive production of waste material that can largely be recycled. Waste in the form of particles is also generated during the production of semi-finished products. This is mainly particles and dust, which must be removed so that they do not affect the production process and the operators.

In the specialist literature, sawdust is characterized as a polydisperse bulk material consisting of coarse and medium-coarse fractions, i.e., a bulk material with grain sizes above 0.3 mm, while the proportion of finer fractions with smaller chip sizes is not excluded. According to the classification indicators of bulk materials stated in STN 26 0070, sawdust is classified as B-45UX i.e., a fine-grained loose mass (0.5 ÷ 3.5 mm) that is hygroscopic, and low-flowing and an abrasive mass with a tendency to clump [2].

Sawdust can be used as a secondary raw material. It is one of the starting raw materials for the production of agglomerated chip materials and the chemical processing of wood, a valuable raw material for energy use by direct combustion, and the basic raw material for the production of dimensionally and energetically homogenized fuel (briquettes and pellets) [3].

The carcinogenic risk to humans posed by occupational exposures to wood dust and formaldehyde needs to be evaluated, since a number of occupational situations that involve exposure to wood dust also entail exposure to formaldehyde, such as in plywood and particleboard manufacturing, furniture- and cabinet-making, and parquet floor sanding and varnishing. The highest occupational exposures were noted to occur in wood furniture

and cabinet manufacturing, especially during machine sanding and similar operations, in the finishing departments of plywood and particleboard mills, and in the workroom air of sawmills and planer mills near chippers, saws, and planers. Citing findings from several recent well-designed case-control studies, this study concludes that occupational exposure to wood dust is causally related to adenocarcinoma of the nasal cavities and paranasal sinuses [4–6].

Wood processing into a final product is a very complex technological process. The main aim of wood processing is to create a workpiece with the required shape, dimensions, and surface quality [7]. One of the most-used methods of woodworking is milling [8–13]. The quality of the processed surface by milling is affected by various factors, such as the cutting conditions, the blunting of the tool, and the appropriately chosen tool [14–18]. In the case of the wear of the tool during a long period of milling, the vibration frequency may increase, resulting in a decrease in the quality of the milled surface. Tool wear is affected by many factors including the workpiece material, cutting parameters, tool geometry and materials, tool temperature, and cooling methods. All these parameters affect the service life of the tool [19–21].

Currently, the chipboard production is a priority direction in the development of the woodworking industry. Particle board (chipboard) is a material used in the production of cabinet furniture and construction. The popularity of chipboard is also due to the fact that manufacturers of this board material are trying to introduce the promising developments of scientists, to keep up with the times [22,23]. The technology for the production of particle boards is a complex process including a number of important operations. The quality indicators of the finished product largely depend on it. For this reason, we chose this material as the material for our experiment. In addition to the purely technological aspects, the environmental safety of chipboard production is the most relevant, which is reflected in the modern patent, scientific, and technical literature. Chipboard is composed of particles and thin slivers of wood that are made by cutting the wood feedstock with rotating knives and shearing the wood into small elements. The characteristics of chipboard are its low cost, its high thickness, and the capability to manufacture large-dimension boards. Chipboard manufactured from waste materials has an extra carbon offset value, making a contribution to a sustainable environment.

Particleboard is a composite panel product consisting of cellulosic particles of various sizes that are bonded together with a synthetic resin or binder under heat and pressure. Particle geometry, resin levels, board density, and manufacturing processes may be modified to produce products suitable for specific end uses. At the time of manufacture, additives can be incorporated to impart specific performance enhancements including greater dimensional stability, increased fire retardancy, and moisture resistance.

Today's particleboard gives industrial users the consistent quality and design flexibility needed for fast, efficient production lines and quality consumer products. Particleboard panels are manufactured in a variety of dimensions and with a wide range of physical properties that provides maximum design flexibility for specifiers and end users.

## 2. Materials and Methods

### 2.1. CNC Machine

The experiments were carried out on a 5-axis CNC machining center SCM Tech Z5 manufactured by the company SCM Group, Rimini, Italy.

The basic technical parameters of the machining center given by the manufacturer are provided in Table 1.

**Table 1.** Technical and technological parameters of CNC machining center SCM Tech Z5.

| Parameter | Range |
|---|---|
| Userful desktop | 3050 × 1300 × 300 mm |
| X-axis speed | 0–70 m·min$^{-1}$ |
| Y-axis speed | 0–40 m·min$^{-1}$ |
| Z-axis speed | 0–15 m·min$^{-1}$ |
| Vector rate | 0–83 m·min$^{-1}$ |
| Revolutions | 600–24,000 rpm |
| Power | 11 kW |
| Maximum tool diameter | D = 160 mm |
| Maximum tool length | L = 180 mm |

### 2.2. Tool Parameters

A diamond shank cutter tool with two rows of cutting diamond blades (Diamond Router Cutter Economic Z2 + 1 − D18 × 26L85S = 20 × 50) was used, manufactured by IGM Tools and Machines (Figure 1). The basic technical and technological parameters given by the manufacturer are provided in Table 2. This tool was chosen for its frequent usage in small woodworking companies due to its high tool lifetime and relatively low cost [24–26]. The cutter was used in previous experiments. The usage time was approximately 120 min.

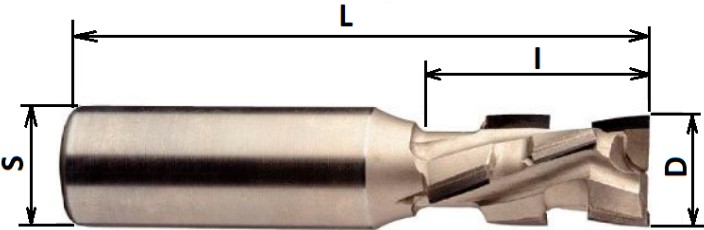

**Figure 1.** Diamond Shank Cutter Economic Z2 + 1 − D18 × 26L85.

**Table 2.** Technical parameters of the milling tool.

| Name | Working Diameter D (mm) | Working Length L (mm) | Diameter of Chucking Shank S (mm) | Number of Cutting Blades | Material of Cutting Edges |
|---|---|---|---|---|---|
| Router Cutter Economic Z2 + 1 | 18 | 26 | 20 | 2 + 1 HW | Diamond |

### 2.3. Milling Wood Samples

A pressed chipboard was used as a sample for milling. The sample had a raw surface without processing, moisture content of 9.5%, and panel density of 600–640 kg·m$^3$. Samples of particleboard blanks with the following dimensions, thickness t = 18 mm, width w = 300 mm, and length l = 500 mm, were used in the experiment. The specimens were machined by cylindrical, circumferential milling through the entire thickness, with a diamond shank milling cutter with the following technological parameters: constant depth of cut e = 4 mm; rotation speed of spindle with cutting tool n = 18.000 rpm; feed speed vf = 4, 6, and 8 m·min$^{-1}$. For each combination of parameters, six specimens in total were collected. The conventional milling (up-milling) method was used for the experiment.

The sawdust obtained during milling was then scanned using a Nikon D5200 camera. This camera was placed on a tripod above the scanned area. The shooting lens was a standard camera lens, Nikon AF-S Nikkor 18–55 mm f/3.5–5.6 GDX VR II (Nikon, Bangkog, Thailand). This lens is designed for use with Nikon's DX-format single-lens reflex cameras. A 3× zoom covers the commonly used focal length range of 18–55 mm and a Silent Wave Motor (SWM) from Nikon offers quiet autofocus. Its view angle is 76–28°50′.

During scanning of the measured sawdust, particles may overlap each other. In this case, the overlapping sawdust would be evaluated as one particle, which would introduce an error into the measurement. So that the sawdust in the sample does not overlap, the particles are separated from each other during the scanning itself using a vibrating table (Figure 2).

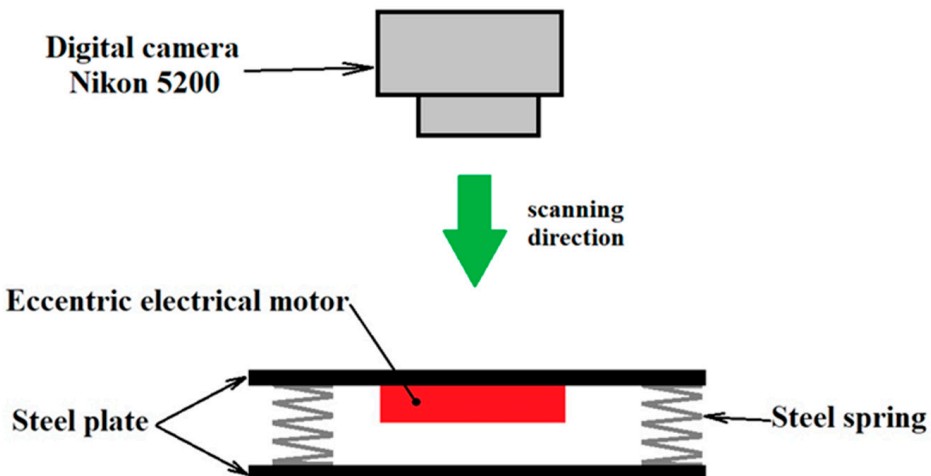

**Figure 2.** Vibrating table for separating wood particles.

The vibrating table was assembled from two steel plates, which are connected by four springs. The springs were placed in the corners of the plates and fixed by welding. On the bottom of the upper plate, there is an eccentric electric motor in the middle, the movement of which creates an oscillating movement by the upper plate. The speed of the motor and, thus, the strength of the vibrations are adjusted by regulating the supply voltage for the motor. A simple circuit with an LM317 regulator was used as a voltage regulator (Figure 3).

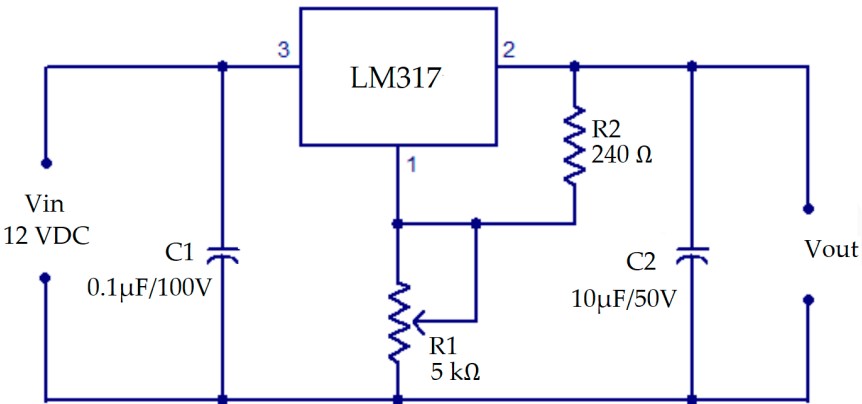

**Figure 3.** Circuit for voltage regulation. LM317 pins: 1: Adjust; 2: $V_{OUT}$; 3: $V_{IN}$.

LM317 is a monolithic integrated circuit in TO-220 packages intended for use as a positive adjustable voltage regulator. It is designed to supply more than 1.5 A of load current, with an output voltage adjustable over a range from 1.2 to 37 V. The nominal output voltage is selected by means of a resistive divider, making the device exceptionally easy to use and eliminating the stocking of many fixed regulators. The input voltage for the controller was 12 VDC voltage from the main adapter. The output voltage from the regulator ranged from 1.25 to 11.3 V. This voltage powered the eccentric motor in the vibrating table.

The particles were scanned with the following parameters (Table 3):

**Table 3.** Shooting parameters.

| Parameter | Value |
|---|---|
| ISO sensitivity | 100 |
| Shutter speed | 6.0 s |
| Aperture | f/5.6 |
| Focal length | 55 mm |
| Effective pixels | 24.2 Mpix |
| Sensor format | APS-C |
| Image sensor type | CMOS |

Scanning of the samples was carried out in low light so that the shadow of the particles was not visible. Therefore, images were recorded with a long exposure, 6 to 15 s.

A sample of an image with analyzed particles is shown in Figure 4.

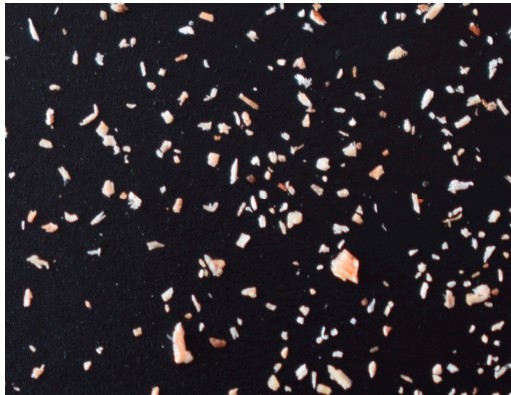

**Figure 4.** Image with analyzed particles.

Images of sawdust taken in this way were subsequently analyzed in the MATLAB program (MathWorks, Natick, MA, USA), using the proposed program (Figure 5).

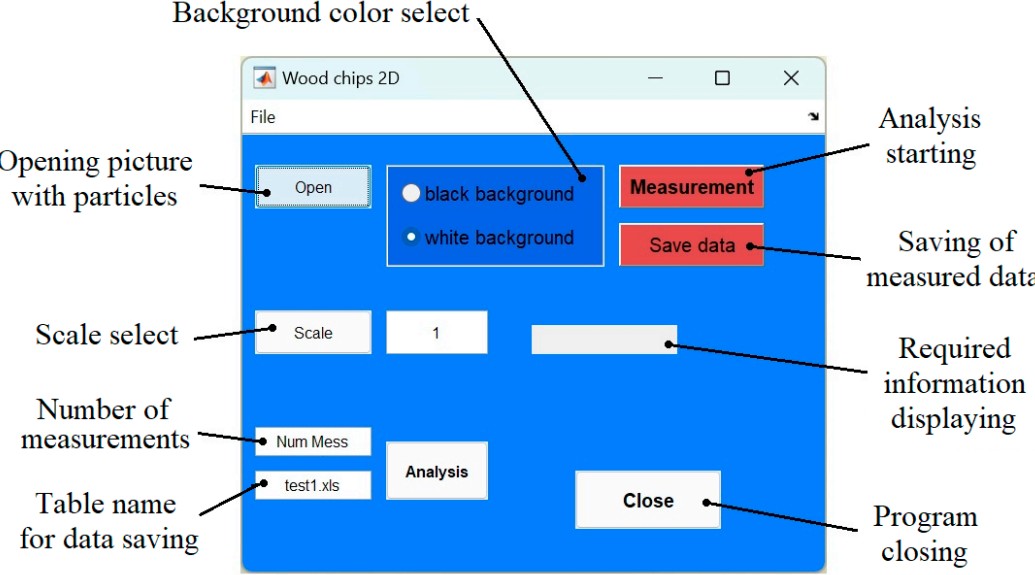

**Figure 5.** Proposed MATLAB program for wood particles' analysis.

The text block of the displayed required information was mainly used for the design of the application for listing certain information, for example, the value of the content of

the test object. During the experiment, the total number of found objects in the currently analyzed image was written into this block.

The "Analysis" button was designed for quick analysis of measured data, such as a histogram of detected particle areas. However, it was not used in the experiment; all analyses were performed in the program Statistica (TIBCO Software Inc., Arlington, VA, USA).

The program works according to the following algorithm (Figure 6).

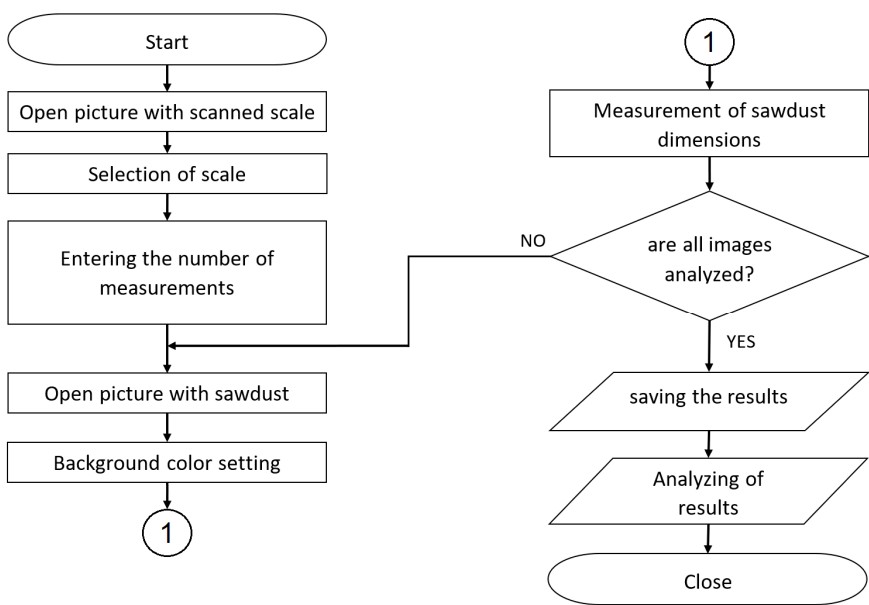

**Figure 6.** Program algorithm.

First, an image with a reference distance is opened, which helps to determine the scale of the conversion from digital image to the metric system. This step is important to accurately determine the dimensions. The MATLAB program detects all dimensions from the digital image; therefore, the measured values are in pixels. For the conversion to metric dimensions, a conversion coefficient is found. To calculate it, a known distance is scanned in the first opened photo. In the experiment, an office ruler was used. In this ruler, 2 points with a known distance were selected by clicking the mouse. When clicked, the X Y coordinates of the given points were determined, and the distance between the points in pixels was calculated using relationship (1).

$$Dist_{Px} = \sqrt{(X_2 - X_1)^2 + (Y_2 - Y_1)^2} \tag{1}$$

where

$Dist_{Px}$—distance between the selected points in pixels;
$X_1$, $Y_1$—coordinates of the first selected point;
$X_2$, $Y_2$—coordinates of the second selected point.

From this distance in pixels in the image and from the known distance on the ruler, the conversion coefficient is then calculated according to relationship (2):

$$Con.Coef = \frac{Dist_{mm}}{Dist_{Px}} \tag{2}$$

where

*Con. Coef*—conversion coefficient;
$Dist_{mm}$—known distance in metric system (mm);
$Dist_{Px}$—known distance in pixels.

Subsequently, the analyzed image is opened, in which it is necessary to select the background color of the sawdust, which depends on whether there are light wood particles on a black background or dark wood particles on a white background. These combinations are suitable for the contrast between the searched sawdust and the background, so that they can be easily identified. In the case of a background with a similar color to the searched objects, particles may be incorrectly assigned to the background, or false objects may be created [27,28].

For next analysis, this image is converted into binary form using a function:

$$im2bw(I, \text{graythresh}(I)) \tag{3}$$

where *I*—a variable representing the loaded image.

Function im2bw converts the input image to a binary form, in which the pixels belonging to the sawdust have the value of 1 (white), and the other pixels have a value of 0 (black). The decision level for this transfer is calculated using a function: graythresh(I). This computes a global threshold T from grayscale image I, using Otsu's method. Otsu's method chooses a threshold that minimizes the intraclass variance of the thresholded black and white pixels [27–29]. During this binarization, fictitious holes may be created, due to the structure of the sawdust. These are subsequently removed using a function:

$$\text{imfill}(BW,'\text{holes}') \tag{4}$$

where

BW—input binary image;
'holes'—parameter of the imfill function.

Function imfill(BW,'holes') fills holes in the input binary image BW. Using parameter 'holes', only holes in objects are removed (Figure 7). Hole is a set of background pixels that cannot be reached by filling in the background from the edge of the image.

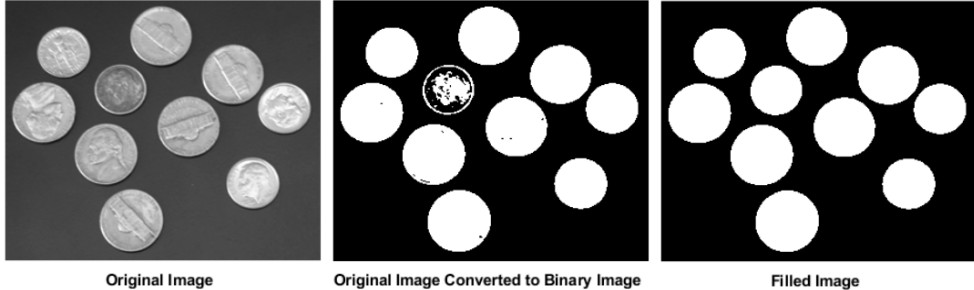

**Figure 7.** Removing holes in objects [30,31].

In a binary image modified this way, the dimensional characteristics of the sawdust are subsequently detected by pressing the "Measurement" button.

It is possible to determine the dimensional parameters of the sawdust in the digital image modified in this way. The following functions were used in the MATLAB program to determine sawdust parameters:

$$\begin{array}{c} \text{regionprops}(BW, \text{properties}) \\ \text{bwferet}(BW, \text{properties}) \end{array} \tag{5}$$

where

BW—input binary image;
properties—specified, required calculated properties.

Using the regionprops function, the required properties of the found particles are calculated. The list of these characteristics is specified as Properties in the function region-

props. To measure the dimensions of the sawdust, the following were determined: Area, Perimeter, Centroid, Orientation, and Circularity.

Function bwferet measures the Feret properties of objects in an image and returns the measurements in a table. The input properties specify the Feret properties to be measured for each object in input binary image BW. The measured Feret properties include the major and minor axis length, Feret angles, and endpoint coordinates of Feret diameters.

The Feret properties of an object are measured by using boundary points on the antipodal vertices of the convex hull that encloses that object (Figure 8).

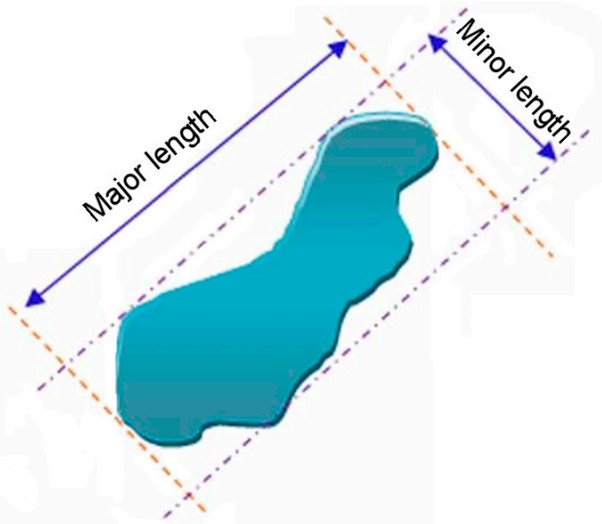

**Figure 8.** Major and minor length.

Area of individual sawdust is determined with a parameter 'Area'. This parameter counts all pixels belonging to individual sawdust in the binary image.

The Perimeter measurement of sawdust is determined using a parameter 'Perimeter'. Function regionprops computes the perimeter by calculating the distance between each adjoining pair of pixels around the border of the region.

The position of the center of sawdust is determined by a parameter 'Centroid', which detects the horizontal and vertical coordinates of the position of the center of particle in the image.

The rotation of sawdust in the image is detected by a parameter 'Orientation'. This represents an angle between the x-axis and the major axis of the ellipse that has the same second moments as the region, returned as a scalar. The value is in degrees, ranging from −90° to 90° [27,28].

Roundness of objects is returned as a structure with parameter 'Circularity'. The structure contains the circularity value for each object in the input image. The circularity value is computed as

$$\frac{4 \cdot Area \cdot \pi}{Perimeter^2} \tag{6}$$

Since MATLAB detects dimensional information about found objects in pixels, the obtained information is converted to metric system. This is accomplished by multiplying the perimeter and the min and max dimension data by the conversion coefficient that was calculated at the beginning of the measurement. Particle area data are multiplied by the square of the coefficient. The circularity parameter is not recalculated by this coefficient because it is a relative quantity. The measured data are sent to an Excel table. The data modified in this way are saved in an Excel table using the "xlswrite" function. The data can be further processed in the Excel program. We exported these data to the program Statistica.

## 3. Results

The proposed program allows for the measurement of the dimensions of each individual particle. A sample of the determined dimensions is displayed in Figure 9.

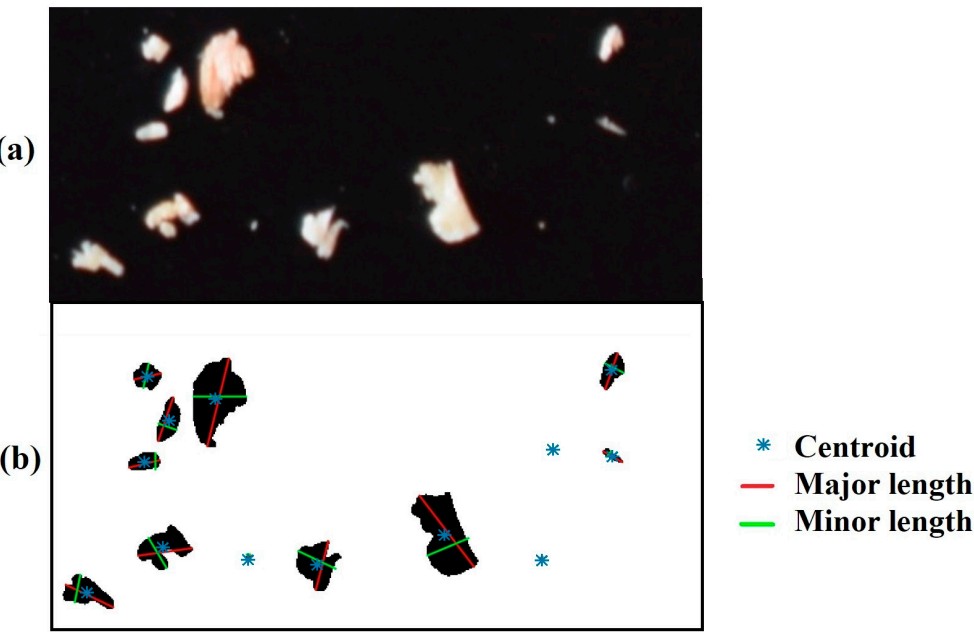

**Figure 9.** Section from the analyzed image with sawdust: (**a**) original image; (**b**) analyzed image.

The found dimensions were analyzed by one-way ANOVA in the Statistica program (TIBCO Software Inc., USA). Figure 10 shows the weighted means of the area of the analyzed sawdust for individual feed speeds.

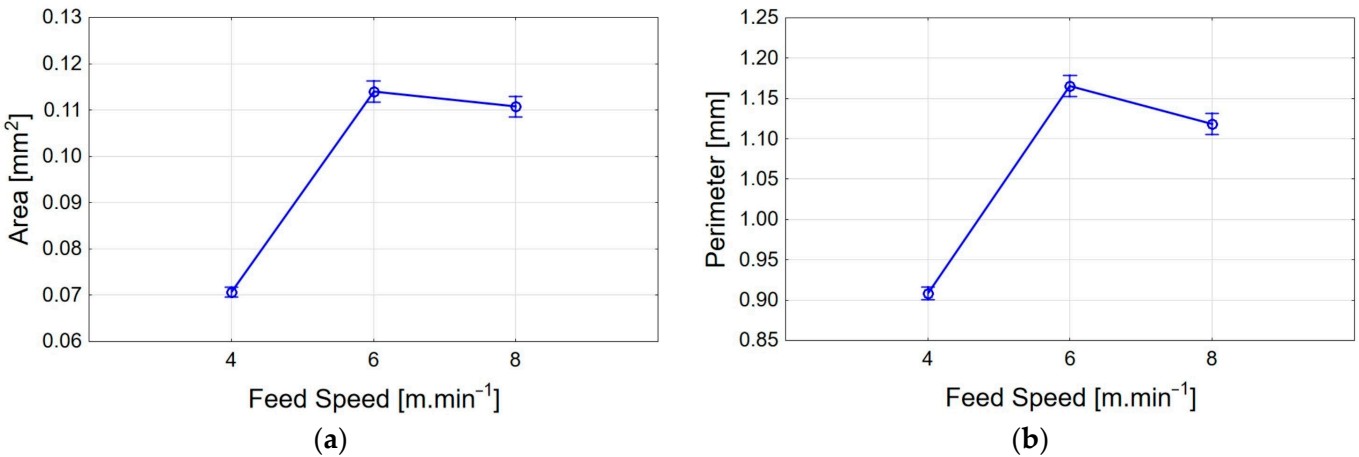

**Figure 10.** Weighted mean of dimensional characteristics of the analyzed sawdust: (**a**) sawdust area; (**b**) sawdust perimeter.

As shown in Figure 10, the area and perimeter of the sawdust are changing with different feed speeds. The smallest particles were formed by milling with the smallest feed speed. By increasing the feed speed, the size of the generated sawdust also increased. The largest sawdust was created at a feed speed of $v_f = 6$ m·min$^{-1}$.

The weighted means of the major and minor axes are shown in Figure 11.

To measure by area and perimeter, the major and minor axis dimensions were recorded at a feed speed of 4 m·min$^{-1}$. At higher feed speeds, the dimensions were larger. The largest sawdust dimensions were recorded at a feed speed of 6 m·min$^{-1}$.

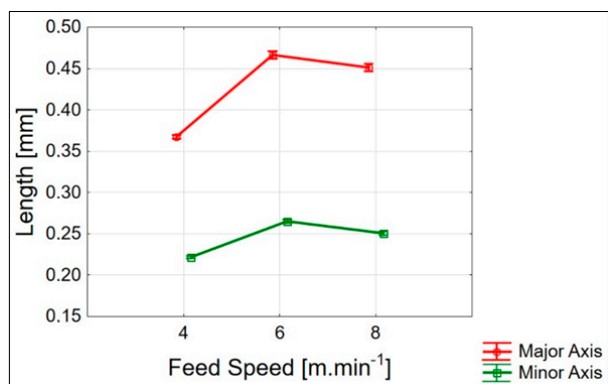

**Figure 11.** Weighted mean of the largest and smallest dimensions of the analyzed sawdust.

The statistical significance of the detected parameters in the change in speed was determined using Duncan's test (Table 4).

**Table 4.** Duncan's test of areas.

| Feed Speed | {1} | {2} | {3} |
|---|---|---|---|
| 4 m·min$^{-1}$ | | 0.000011 | 0.000009 |
| 6 m·min$^{-1}$ | 0.000011 | | 0.011701 |
| 8 m·min$^{-1}$ | 0.000009 | 0.011701 | |

Table 3 shows that the change in feed speed is statistically significant because the probability of the similarity of the datasets is less than 5%.

Using the described method of determining the sawdust dimensions, it is also possible to determine the roundness of objects (Circularity). For a perfect circle, the circularity value is 1.

As shown in Figure 12, the shape of the sawdust is similar to a circle in the sample. For small particles, however, the circularity increases, which is due to the fact that these small particles have a needle-like shape. They have a small area but a larger perimeter. For larger particles, the roundness is smaller because these particles have a shape similar to a circle. The analysis of the variance of the circularity is shown in Figure 13.

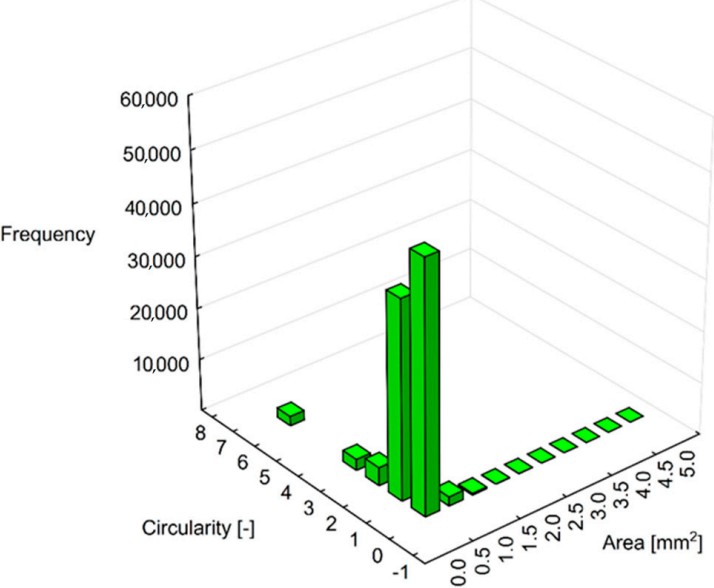

**Figure 12.** Circularity of sawdust in different areas.

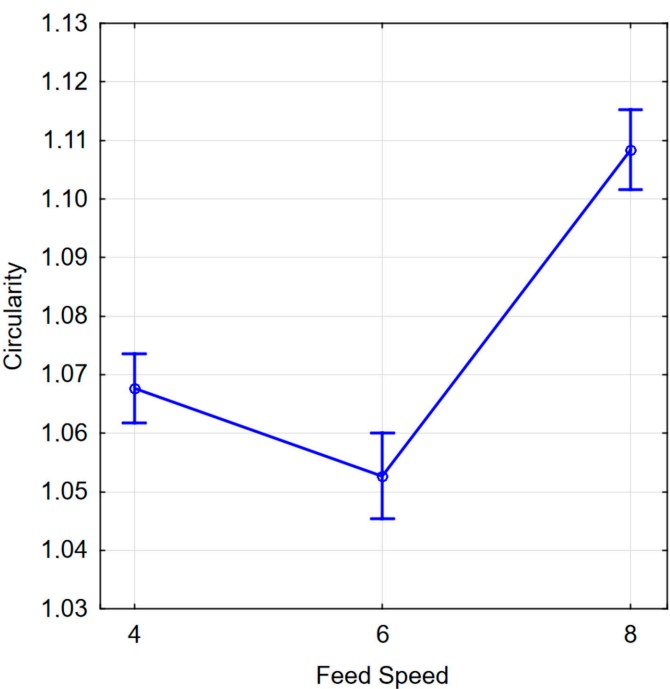

**Figure 13.** Analysis of variance of circularity.

For a comparison with the sieve analysis, the obtained results were converted from the percentage representation of the sawdust to an area corresponding to the sieves with fractions: 2, 1, 0.5, 0.25, 0.125, 0.063, 0.032, and less than 0.032 mm. Figure 14 shows the results.

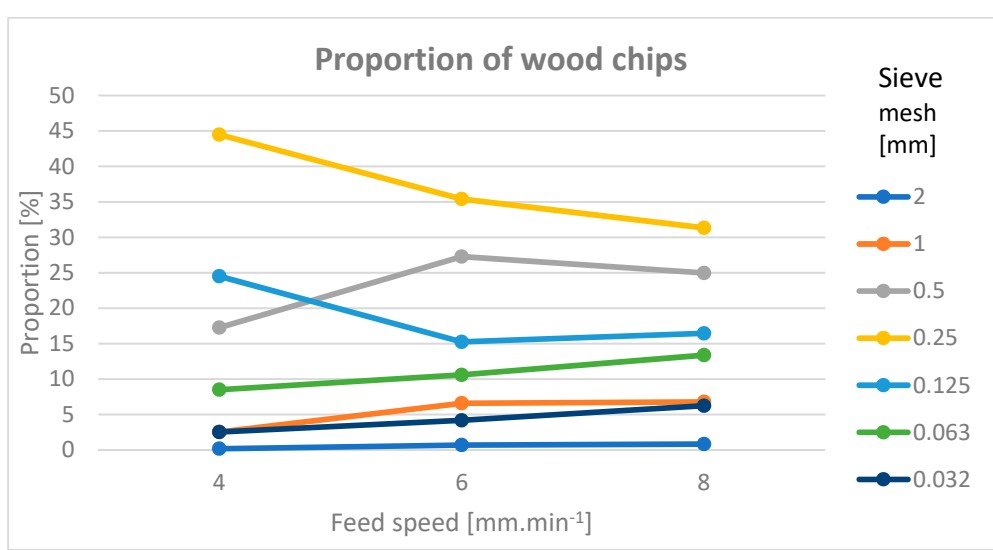

**Figure 14.** The percentage of individual sawdust fractions.

Figure 14 shows that the largest share of the sawdust was from the fractions 0.25 and 0.5 mm. The major factions (2 a 1 mm) had a small share, approximately 5% at each feed speed.

## 4. Discussion

An optical analysis of the sawdust and other small materials encounters the problem of overlapping particles during scanning. When several particles overlap each other, such a cluster is typically identified as one separate particle. This problem can be solved using a

vibrating table, on which the particles of re ias separated from each other using vibrations before scanning.

Another problem can be capturing all the particles for analysis. The particles' removal efficiency during CNC machining of the particleboard highly depends on the operation type realized by the CNC center as well as the machining pathway [32,33]. For pocketing, the quantity of particles that remained on the panel after routing was negligible and, therefore, the exhaustion efficiency was near 100%. For milling, however, it was at the level of 87%, which was not satisfying. In this study, a particle size analysis of the sawdust was also performed. It showed that the wood particles left over on the machine and around it were not smaller than 0.1 mm. The efficiency of the wood particles' removal decreased with the particles' size increase.

It is possible to identify the effect of shear force on the proportion of the smaller particles within the fine fractions in terms of the influence of the physical and chemical properties of sawed and sanded material, as well as the shape, the dimensions, the sharpness of cutting tools, and technological factors. A feed rate reduction means a decrease in the nominal thickness of the particle, and, thus, the particles move between finer fractions. This fact was also confirmed by other studies [34–36]. The formation of dust particles can occur in all open places of machines as well, especially on the premises of CNC machines as a result of maintenance, repairs, cleaning, inspections, tool changes, etc. [30,31,37–39].

This paper is based on the standard scientific methodologies for the evaluation of particles from the wood milling process, which are accepted for their scientific capacities, but, at the same time, we consider it necessary to discuss this topic from the point of view of objectivity and in the context of the stated findings.

As Kminiak (2021) wrote, it is very difficult to determine the content of the finest dust particles. This content may not be captured by the camera due to the complicated shape of the particles, leading to a possibly incomplete data analysis. Therefore, a smaller focal length camera or a microscope should be used for wood dust with a larger dimensional span. Only then is it possible to detect and quantify the content of the finest dust particles and, thus, to estimate the occupational health risks accordingly [40,41].

One of the ways to improve the chips' geometric measurement is to use the optical method, which was proposed by Sandak et al. (2005) and also by Palubicki et al. (2007). This method has many advantages, since it is simple and fast, does not use very expensive equipment, and has high accuracy [42,43].

## 5. Conclusions

This research demonstrated the possibility of a more complex analysis of sawdust using the proposed program. In the commonly used methods, for example sieve analyses, the result is only the percentage representation of the size of the individual fractions compared to the total sample [44,45]; thus, the described method allows for obtaining more information about the measured sawdust sample. The dimensional characteristics are determined for each individual particle.

During the analysis of the particles generated during the milling process at different feed speeds, it was found that the smallest particles are generated at a feed speed of $4 \text{ m·min}^{-1}$. The largest sawdust particles were generated at a feed speed of $6 \text{ m·min}^{-1}$. In order to reduce health risks during milling, it is, therefore, not advisable to use low feed speeds. They produce smaller chips and dust that could endanger the health of the operating personnel.

The formation of fine wood dust particles represents a significant occupational hazard to the health and safety of workers. The results obtained can be used for optimizing the technological programs of CNC milling machines, thus reducing the occupational exposure to harmful wood dust emissions in the wood processing industry.

The improvement of the work environment in wood processing and furniture enterprises, by adopting adequate occupational safety and health practices, is desirable not only from the perspective of workers but also because it contributes substantially to labor pro-

ductivity by enhancing workers' motivation, increasing competitiveness, and promoting economic growth.

During particle scanning, this method was found to be quite time-consuming. This was mainly due to the dark contrasting background of the sawdust. For further experiments, it would be more appropriate to provide light under the sawdust, which would speed up the scanning. Moreover, the particles illuminated in this way would have sharper contours.

In this experiment, a standard lens for a Nikon camera was used. Its maximum focal length is 55 mm. For further research, we plan to try other lenses with longer focal lengths as well as other cameras with extra-long focal lengths. A longer focal length allows for shooting at a smaller view angle. Therefore, the investigated particles scanned in this way should have clearer details.

**Author Contributions:** Conceptualization, P.K. (Pavol Koleda) and P.K. (Peter Koleda); methodology, P.K. (Pavol Koleda); software, P.K. (Pavol Koleda); validation, P.K. (Pavol Koleda), P.K. (Peter Koleda), and M.H.; formal analysis, P.K. (Pavol Koleda) and M.H.; investigation, P.K. (Pavol Koleda); resources, M.H.; data curation, M.J.; writing—original draft preparation, P.K. (Pavol Koleda); writing—review and editing, P.K. (Pavol Koleda) and Á.H.; visualization, P.K. (Peter Koleda); supervision, P.K. (Pavol Koleda); project administration, P.K. (Pavol Koleda); funding acquisition, P.K. (Peter Koleda). All authors have read and agreed to the published version of the manuscript.

**Funding:** This research was funded by grant number VEGA 1/0791/21.

**Data Availability Statement:** Not applicable.

**Acknowledgments:** This publication is the result of the following projects' implementation: project VEGA 1/0791/21, "Research of non-contact method of analysis of small and dust particles arising in the production process with a prediction of negative effects of dust particles" and project APVV-20-0403, "FMA analysis of potential signals suitable for adaptive control of nesting strategies for milling wood-based agglomerates"; thanks go to the support under the Operational Program Integrated Infrastructure for project II–NITT SK II, "National infrastructure for supporting technology transfer in Slovakia", co-financed by the European Regional Development Fund.

**Conflicts of Interest:** The authors declare no conflict of interest.

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
