# Peer review of "Experimental Granulometric Characterization of Wood Particles from CNC Machining of Chipboard"

_applsci, doi:10.3390/app13095484_

Round 1

Reviewer 1 Report

Dear Authors,
in my opinion, your manuscript is up to date due to the topics you have undertaken (small dimensions of dust particles in the human environment) and the methods you have applied (optical methods of wood particle dimensions analysis). However, below, please find several remarks:
- general suggestion: say "particles" to wood pieces that the panel is made of and that are cut during milling; say "chips" to larger wood pieces used in the particleboard industry (between roundwood and particles) to produce particles for panel making
- line 4 - it seems like you have two affiliations only instead of 4, as said in the last author's name
- line 14 - I'm afraid you are not allowed to say that you determined the "mass fraction" of sawdust, since you did not measure the weight of the fractions; probably you just can say about size fractions
- lines 29 and 33 - I suggest unifying the spelling of the names "particle board" and "particle-board" to "particleboard"
- to Introduction section: since you have analysed the relation between machining parameters and sawdust dimensions measured optically, I suggest saying more in the Introduction section about the state of the art regarding machining and sawdust properties
- Figure 1 can be removed since it does not provide any significant information needed to understand and/or repeat the research
- lines 60-61 - the sentence "A diamond shank cutter with two cutting blades was used for the experiment." can be removed since similar information is provided above in the same paragraph
- the "Tool parameters" paragraph - what was the status of the tool? New? Used?
- the "Milling wood samples" paragraph: what was panel density? Panel surface (raw or finished/laminated)? Moisture content?
- when describing Figure 4, I suggest avoiding the word "application" since what you presented is only a user interface, and hard to call it an "application". More useful would be the algorithm scheme
- Figures 7, 8 and 10 (plots) and also in the text - should be dots instead of commas in decimals
- lines 176 and 177 - I'm not sure that sentence is proper: "For small particles, however, the circularity increases, which is due to the fact that these small particles have a needle-like shape". In my opinion, the needle shape is far away from the circular shape...
- in the Discussion section I suggest referring your finding to the following open-access paper, which contains a similar technique of particle measuring: Palubicki B., Kowaluk G., Frackowiak I. (2007): Convexity - additional parameter to chips geometry characterization. DREWNO-WOOD 2007, vol. 51, no. 178
- lines 214-219 - when saying about the difficulties in the measurement of fine particles (dust), please refer to Tomasz Rogozinski, Poznan, Poland, achievements, where the application of laser technology is described
- line 301 - improper spelling of Palubicki (line 326 is OK)
Best regards!

Author Response

Response to Reviewer 1 Comments

Point 1: General suggestion: say "particles" to wood pieces that the panel is made of and that are cut during milling; say "chips" to larger wood pieces used in the particleboard industry (between roundwood and particles) to produce particles for panel making.

Response 1: I unified the naming of particles in manuscript.

Point 2: Line 4 - it seems like you have two affiliations only instead of 4, as said in the last author's name

Response 2: I didn't notice the original numbering, I edited it in manuscript.

Point 3: Line 14 - I'm afraid you are not allowed to say that you determined the "mass fraction" of sawdust, since you did not measure the weight of the fractions; probably you just can say about size fractions

Response 3: I edited the term “mass” it in manuscript.

Point 4: Lines 29 and 33 - I suggest unifying the spelling of the names "particle board" and "particle-board" to "particleboard"

Response 4: I unified the term “particleboard”.

Point 5: To Introduction section: since you have analysed the relation between machining parameters and sawdust dimensions measured optically, I suggest saying more in the Introduction section about the state of the art regarding machining and sawdust properties

Response 5: To the Introduction I added the mentioned parts.

Point 6: Figure 1 can be removed since it does not provide any significant information needed to understand and/or repeat the research

Response 6: I deleted the picture 1.

Point 7: Lines 60-61 - the sentence "A diamond shank cutter with two cutting blades was used for the experiment." can be removed since similar information is provided above in the same paragraph

Response 7: I removed the mentioned sentence.

Point 8: The "Tool parameters" paragraph - what was the status of the tool? New? Used?

Response 8: I added the period of use of the cutter (120 min).

Point 9: The "Milling wood samples" paragraph: what was panel density? Panel surface (raw or finished/laminated)? Moisture content?

Response 9: I added information about the sample to the section “Milling wood samples”.

Point 10: When describing Figure 4, I suggest avoiding the word "application" since what you presented is only a user interface, and hard to call it an "application". More useful would be the algorithm scheme

Response 10: I replaced the term “application” with the term “program”. I added the algorithm of the program work.

Point 11: Figures 7, 8 and 10 (plots) and also in the text - should be dots instead of commas in decimals

Response 11: I have set decimals in the images and in the text to dots.

Point 12: Lines 176 and 177 - I'm not sure that sentence is proper: "For small particles, however, the circularity increases, which is due to the fact that these small particles have a needle-like shape". In my opinion, the needle shape is far away from the circular shape...

Response 12: In the matlab program, the circularity parameter is calculated from the perimeter and area of the particle by formula “perimeter2 / (4*π*area)”. For small particles with a needle-like shape, which have a small surface area but a larger circumference, a large value is calculated in this way. Maybe this parameter is not so suitable for describing sawdust.

Point 13: In the Discussion section I suggest referring your finding to the following open-access paper, which contains a similar technique of particle measuring: Palubicki B., Kowaluk G., Frackowiak I. (2007): Convexity - additional parameter to chips geometry characterization. DREWNO-WOOD 2007, vol. 51, no. 178

Response 13: I added the mentioned article to the discussion

Point 14: Lines 214-219 - when saying about the difficulties in the measurement of fine particles (dust), please refer to Tomasz Rogozinski, Poznan, Poland, achievements, where the application of laser technology is described

Response 14: Thanks for the contact, I will contact him for further research.

Point 15: Line 301 - improper spelling of Palubicki (line 326 is OK)

Response 15: I corrected the mistake in the name.

Reviewer 2 Report

please see tha attached file for the comments.

Author Response

Response to Reviewer 2 Comments

Point 1: Selection of proposed material and the targeted industry is smaller in scope and application; A research work must be broader in scope and targets.

Response 1: I expanded the article gradually in individual parts

Point 2: What are the technological benefits of the study, Is it applicable to industries other than furniture industry?

Response 2: The proposed program for determining particle sizes can of course also be used for other materials from other areas of industry. The study predicts the inappropriateness of using low feed rates in circumferential milling.

Point 3: In the line 43, tool wear is mentioned, did tool wear happens when work piece is much softer than the tool (which is very hard in nature).

Response 3: The wear was meant as a manifestation of long-term use of the tool, not during the experiment. I also edited the text in this sense.

Point 4: Mention the depth of cut materials and method section. Also mention it in table 1.

Response 4: I added the milling depth.

Point 5: Mention material of the blades in line 57.

Response 5: I added the material of the blades. The manufacturer does not state the exact composition of the material. These are commercially sold cutters.

Point 6: Improve the overall English in the paper, check spellings and adverbs specially in lines 112,115,120,214, and 222 etc.

Response 6: I edited the english in the paper.

Point 7: Improve the image quality of figure 4.

Response 7: I adjusted the quality of the mentioned image.

Point 8: The properties mentioned in line 123 are interdependent to each other, please quantify the dependence separately for each.

Response 8: I added circularity dependencies to the results, other properties are quantify on the previous graphs.

Point 9: What is the validation of the statistical significance of the data.

Response 9: Other authors also reached the same results when analyzing chipboard milling.

Point 10:  Mention the significance of the difference of dust particle size in line 227.

Response 10: I added the risks of milling with a small feed speed, when smaller particles are formed.

Point 11: In conclusion section, lines 230-234 are not required.

Response 11: I removed these lines.

Round 2

Reviewer 1 Report

Dear Authors,
thank you for your careful and detailed response to the remarks mentioned in the review.
No other remarks from my side.
Best regards!

Reviewer 2 Report

the authors have removed my concerns.